

# Atlantic Hurricane response to Sahara greening and reduced dust emissions during the mid-Holocene

Samuel Dandoy[1], Francesco S.R. Pausata[1], Suzana J. Camargo[2], René Laprise[1], Katja Winger[1], and Kerry Emanuel[3]

[1]Department of Earth and Atmospheric Sciences, Université du Québec à Montréal
[2]Lamont-Doherty Earth Observatory, Columbia University, Palisades, NY, USA
Massachusetts Institute of Technology, Cambridge, MA,
[3]Program in Atmospheres, Oceans, and Climate, Massachusetts Institute of Technology, Cambridge, Massachusetts, USA

*Correspondence to*: Samuel Dandoy (dandoii@hotmail.ca)

**Abstract.** We use a high-resolution regional climate model to investigate the changes in Atlantic tropical cyclone (TC) activity during a warm climate state, the mid-Holocene (MH: 6,000 yrs BP). This period was characterized by increased boreal summer insolation, a vegetated Sahara, and reduced airborne dust concentrations. A set of sensitivity experiments were conducted in which solar insolation, vegetation and dust concentrations were changed in turn to disentangle their impacts on TC activity. Results show that the greening of the Sahara and reduced dust loadings ($MH_{GS+RD}$) lead to a larger increase in the number of Atlantic TCs (27% ) relative to the pre-industrial climate (PI) than the orbital forcing alone ($MH_{PMIP}$; 9%). The TC seasonality is also highly modified in the MH climate, showing a decrease in TC activity during the beginning of the hurricane season (June to August), with a shift of its maximum towards October and November in the $MH_{GS+RD}$ experiment relative to PI. MH experiments simulate stronger hurricanes compared to PI, similar to future projections. Moreover, they suggest longer lasting cyclones relative to PI. Our results also show that changes in the African Easterly Waves are not relevant in altering the frequency and intensity of TCs, but they may shift the location of their genesis. This work highlights the importance of considering vegetation and dust changes over the Sahara region when investigating TC activity under a different climate state.

## 1 Introduction

Tropical Cyclones (TCs) are one of the most powerful atmospheric phenomena on Earth. Every year, billions of dollars are lost due to damages caused by these natural disasters on the environment leaving affected population to struggle with large floods and vast quarters to rebuilt (Villarini et al., 2014). According to the National Oceanic and Atmospheric Administration (NOAA) National Climatic Data Center, the frequency of extreme events exceeding one billion dollars in damages over the US has significantly increased over the last several years. Of all the different natural disasters, TCs are the

most dominant accounting for up to ~45% of the total damages and losses (Smith & Katz, 2013). With rising coastal population and the increasing value of infrastructure (Pielke Jr et al., 2008), future damage loses are liable to increase even



further (Schmidt et al., 2009; Woodruff et al., 2013). In this context, the need for understanding the future projections in TC activity is of paramount importance.

With the growing interest of the population and the scientific community in climate change, many questions still remain concerning the future of TCs intensity and frequency in the Atlantic Ocean in a warmer climate. Some studies based on past observations have shown an intensification of the TCs during the last decades (e.g. Emanuel, 2005) with potentially lower translation speed (Kossin, 2018) and suggest possible future continuation of these trends. Most studies project a decrease in global TC frequency (see Knutson et al., 2019 and reference therein); however, a few indicate an increase (e.g., Emanuel,

2013; Knutson et al., 2020; Vecchi et al., 2019). In particular, Knutson et al. (2013; 2020) and Walsh et al. (2016) point towards higher frequency of the most intense storms in a warmer future climate.  These changes in TC activity may be influenced by a displacement of the Intertropical Convergence Zone (ITCZ) or by the northward extension of the West African Monsoon (WAM) through impacts on the energy fluxes of the global oceanic and atmospheric circulation (Schneider et al., 2014; Seth et al., 2019). Recent studies (Biasutti, 2013; Giannini & Kaplan, 2019) suggest a possible

strengthening of the WAM, with increased rainfall over the Sahel region with a potential greening of this area as well as a reduction of airborne dust emissions (Evan et al., 2016) in the near future.

Dramatic intensifications of the WAM have occurred in the past (Shanahan et al., 2009), the most recent during the early and middle Holocene (MH, 12 000 – 5 000 yrs BP) when the WAM was much stronger and extended further inland than today.

The northward penetration of the WAM led to an expansion of the North African lakes and wetlands, as well as to an extension of Sahelian vegetation into areas that are now desert, giving origin to the so-called "Green Sahara" (e.g., Holmes, 2008; Kowalski et al., 1989; Rohling et al., 2004). Therefore, the MH climate represents a good test case to investigate the TC response to a warmer and greener world.

Paleotempestology records are, however, sparse and most of them only span a few millennia, making it difficult to evaluate TC variability further back than the observational period. Nevertheless, records from western North Atlantic suggest large variations in the frequency of hurricane landfalls during the late Holocene, together with strong positive anomalies in the WAM (Donnelly & Woodruff, 2007; Greer & Swart, 2006; Liu & Fearn, 2000; Toomey et al., 2013).

Only a handful of modeling studies investigating TC changes during the MH are currently available (Korty et al., 2012; Koh & Brierley, 2015; Pausata et al., 2017). Both Korty et al. (2012) and Koh & Brierley (2015) have focused on simulations of the Paleoclimate Modelling Intercomparison Project (PMIP), which only account for the change in orbital forcing and the greenhouse gas concentrations during the MH, assuming preindustrial vegetation cover and dust concentrations (Joussaume et al., 1999; Taylor et al., 2012) (Table 1). These studies do not explicitly simulate the changes in TCs, but rather investigate

how key environmental variables affect TC genesis due to the insolation forcing. Both studies suggest a decrease in Northern





Hemisphere and an increase in Southern Hemisphere TC activity despite the changes in summer insolation forcing. More recently, Pausata et al. (2017) used a statistical thermodynamical downscaling approach (Emanuel et al., 2006, 2008) to generate large number of synthetic TCs and assess their changes during the MH with an enhanced vegetation cover over the Sahara and reduced airborne dust concentrations. Their results suggest a large increase in TC activity worldwide and in

particular in the Atlantic Ocean in the MH climate. However, this kind of downscaling approach does not consider how the TC genesis may have been affected by changes in atmospheric dynamics, such as those associated to the African Easterly Waves (AEWs; Gaetani et al., 2017) that are known to seed TC genesis (Caron & Jones, 2012; Frank & Roundy, 2006; Landsea, 1993; Thorncroft & Hodges, 2001; Patricola et al., 2018). Here, we use the same modeling simulations as in Pausata et al. (2017) to drive a high-resolution regional climate model to investigate the impact of the atmospheric dynamics

changes induced by Saharan vegetation and dust reduction on TC activity during the MH compared to the PI climate. This study will compare the dynamical downscaling results to those obtained with the statistical-dynamical downscaling approach used by Pausata et al. (2017), and how they compare with the findings of Koh & Brierley (2015) and Korty et al. (2012). It will also provide insights into how a potential warmer and greener world could alter future Atlantic TC activity.

The paper is structured as follows. The model description, experimental design and the analytical tools used in the study are presented in Section 2. Section 3 focuses on 1) the model's response to the changes in climate conditions on TC activity, 2) the seasonal distribution of TCs and 3) their intensity. Discussion and conclusions are presented in Section 4.

## 2. Model description and methodology

### 2.1 Models

The simulations analysed by Pausata et al. (2016) and Gaetani et al. (2017) performed with an Earth System Model (EC-Earth version 3.1) at horizontal resolution of 1.125°x1.125° and 62 levels in the vertical for the atmosphere (Hazeleger et al., 2012), will be used to drive a developmental version of the 6th generation Canadian Regional Climate Model (CRCM6; see Girard et al., 2014 and McTaggart-Cowan et al., 2019). The experiments with CRCM6 are carried out on a grid mesh of 0.11°. This high horizontal resolution grid allows to capture many processes that are related to TC genesis and simulate

intense tropical cyclones (Strachan et al., 2013; Walsh et al., 2013; Shaevitz et al. 2014; Camargo & Wing 2016; Kim et al. 2018; Wing et al. 2019). CRCM6 is derived from the Global Environmental Multiscale version 4.8 (GEM4.8), an integrated forecasting and data assimilation system developed by the Recherche en Prévision Numérique (RPN), Meteorological Research Branch (MRB), and the Canadian Meteorological Centre (CMC). GEM4.8 is a fully non-hydrostatic model that uses a semi-implicit, semi-Lagrangian time discretization scheme on a horizontal Arakawa staggered C-grid. It can be run

either as a Global Climate Model (GCM), covering the entire globe, or as a nested regional climate model (RCM). In the RCM configuration, the model uses a hybrid-terrain-following vertical coordinate with 53 levels topping at 10 hPa. For shallow convection, GEM uses the Kuo transient scheme (Bélair al., 2005; Kuo, 1965) and for deep convective processes, it





uses the Kain-Fritsch scheme (Kain & Fritsch, 1990). Finally, CRCM6 is coupled at its lower boundary with the Canadian Land Surface Scheme (CLASS; Verseghy, 2000, 2009) and the lake model Flake (Mironov, 2008; Martynov et al., 2012) to

represent the different surfaces. More details regarding GEM4.8 can be found in Girard et al. (2014). In this study, CRCM6 is integrated on a domain encompassing the Atlantic Ocean from Cape Verde to the North American west coast (~25° W to 120° W and 0° to 45° N; see Fig. 1).

## 2.2 Experimental design

We performed three distinct 30-year long experiments with CRCM6 (see Table 1). The first experiment, the control or

reference case, is a pre-industrial climate (PI) simulation that follows the protocol set by the Paleoclimate Modelling Intercomparison Project (PMIP) and the fifth phase of the Coupled Model Intercomparison Project (CMIP5) (Taylor et al., 2012). Two MH simulations were also performed: in the first one the PMIP protocol is followed, only accounting for changes in the orbital forcing and the greenhouse gases concentrations (MH$_{PMIP}$) relative to the PI. The aim here is to evaluate the effect of the insolation forcing alone on TC activity compared to the reference case. In the second MH

experiment, in addition to the changes in the MH$_{PMIP}$, the Sahara Desert was replaced by evergreen shrub and airborne dust concentrations reduced by up to 80% in the EC-Earth experiment (MH$_{GS+RD}$) relative to PI. Due to those changes in vegetation in the Sahara, the albedo of the region decreased from 0.30 to 0.15 and the leaf-area index increased from 0.2 to 2.6 (for details refer to Pausata et al., 2016 and Gaetani et al., 2017).

## 2.3 Tracking algorithm

In this study, a storm tracking algorithm was developed using a three-step procedure (*Storm identification, Storm tracking and Storm lifetime*) to detect tropical cyclones, following previous studies (Gualdi et al., 2008; Scoccimarro et al., 2011; Walsh et al., 2007). In comparison to most routines, our algorithm performs a double filtering approach similar to that applied in Caron & Jones (2012) to ensure that the genesis and dissipation phases of TCs are well represented and that TCs are not counted twice in case of a temporary decrease intensity followed by a re-strengthening. Looser detection criteria

(with lower values than the standard thresholds values) were first used in order to detect all storm centers, then criteria were enforced to standard values following the literature (strict criteria). Centers that satisfy the strict criteria are then classified as being *strong* centers (*Storm identification*) while the others are classified as *weak* centers. To correctly represent each track (*Storm tracking*), the *strong* and *weak* centers are then paired following two different methods: the *storm history* using a similar approach to that of Sinclair (1997) and the *nearest neighbour* method as in Blender et al. (1997), Blender & Schubert

(2000) and Schubert et al. (1998). Once the storm tracks are defined, the algorithm determines the core of each track as the centers sitting between the first and the last *strong* centers found in the track, thus neglecting the genesis and dissipation phases. This subsection of the track (representing the main TC lifetime) has to satisfy a third set of criteria that reject TCs that do not live long enough, or that do not travel a long enough distance, or that do not reach the strength of a Tropical Storm. If the core of the storm track satisfies all these criteria, the genesis and dissipation phase (represented by the *weak*



centers that occurred before the first and after the last *strong* centers) are added to form the complete storm track. A detailed

description of the storm identification and tracking can be found in the supplementary material.

## 2.4 Potential intensity and genesis indices

Many environmental proxies have been used to link the changes in the dynamical and thermodynamical fields to TC activity.

Here two well-known environmental proxies were adopted, the Potential Intensity ($V_{PI}$) and the Genesis Potential Index

(GPI), to investigate the changes between different climate states in TC achievable intensity and in the areas more prone to

develop TCs, respectively. To calculate the theoretical maximum intensity of TCs given specific environmental conditions,

the $V_{PI}$ formulation includes the sea-surface temperature (SST), the temperature at the level of convective outflow ($T_o$), the

ratio of drag and enthalpy exchange coefficients ($C_k/C_d = 0.9$), and the available potential convective energy difference

between an air parcel lifted from saturation at sea level ($CAPE^*$) at the radius of maximum winds and an air parcel located in

the boundary layer ($CAPE_b$). The formula defined by Emanuel (1995) and updated by Bister & Emanuel (1998, 2002) was

used to account for dissipative heating:

$$V_{PI} = \sqrt{\frac{C_k \cdot SST}{C_d \cdot T_o}(CAPE^* - CAPE_b)} \qquad (1)$$

The Genesis Potential Index (GPI) is an empirical fit of the most important environmental variables known to affect TC

formation. These variables include dynamical (wind shear and absolute vorticity) and thermodynamical (potential intensity

and moist entropy deficit) factors. The first genesis index was introduced by Gray (1975, 1979). Since then, various genesis

indices have been formulated (e.g. Emanuel & Nolan, 2004; Emanuel, 2010; Korty et al., 2012). Here the genesis index

formulation from Korty et al. (2012) was used, which is a modified version of the GPI index described in Emanuel (2010).

This GPI includes the entropy deficit between different atmospheric levels, as the one of Emanuel (2010), with the addition

of the «clipped vorticity» (Tippett et al., 2011):

$$GPI = \frac{a[min(|\eta|, 4\times10^{-5})]^3 [max(V_{PI}-35, 0)]^2}{\chi^{4/3}[25+V_{shear}]^4} \qquad (2)$$

where $V_{PI}$ is the potential intensity, $V_{shear}$ is the wind shear between 250 and 850 hPa levels, $\eta$ is the absolute vorticity at

850hPa, and $a$ is a normalization coefficient. The entropy deficit $\chi$ is defined as:

$$\chi = \frac{s_b - s_m}{s_o^* - s_b} \qquad (3)$$

where $s_b$, $s_m$ and $s_o^*$ represent, respectively, the moist entropies of the boundary layer, middle troposphere, and the saturation

entropy at the sea surface. Other indices have shown similar performances to the GPI, as for example the Tropical Cyclone

Genesis Index - TCGI (Tippett, et al. 2011; Menkes et al. 2011).

## 2.5 Regional model evaluation

To evaluate the CRCM6 performance in simulating tropical cyclones, an additional simulation was carried out using the

ERA-Interim reanalysis as lateral boundary conditions and SSTs, and compared with 25-km ERA5 reanalysis for the period

1980 to 2009 (see Copernicus Climate Change Service (C3S), 2017). Our storm-tracking algorithm was used to detect tropical cyclones in the ERA5 reanalysis data, for validation and to ensure the accuracy of simulated results against the observed track density obtained from the Atlantic «Best Track» dataset (HURDAT2) ( Landsea & Franklin, 2013) for the same period (1980-2009). The model evaluation is presented in *Supplementary Material*. In general, our tracking algorithm captures well the main characteristics of the observed tracks (Fig. A1). However, even considering lower threshold values,

the number of detected TCs in ERA5 data is lower than observations. On the other hand, when comparing the ERA-Interim-driven model-simulated TCs in the simulation against observations, there is a better agreement in the mean track characteristics and number of Atlantic TCs (Fig. A1A). Murakami (2014) and Hodges et al. (2017) also find significant biases in the representation of TCs in various reanalysis datasets using different tracking algorithms. In this study, we use a two-tailed Student $t$ test to determine the statistical significance of changes at the 5% confidence level. The significance of

the changes in TC frequency has been determined using twice the standard error of the mean (~5% confidence level).

The Accumulated Cyclone Energy (ACE; Bell et al., 2000) was also calculated. ACE is defined as the sum of the square of the maximum sustained wind speed higher than 35kt every 6h over all the storm tracks. ACE is an integrated measure depending on TC number, intensity and duration, and less sensitive than TC counts to tracking schemes and thresholds

(Zarzycki & Ullrich, 2017). Besides the total ACE, the mean ACE per storm was also considered, and ACE per mean storm duration, in order to analyze the contribution of the different components of ACE to the total value.

To evaluate the statistical significance in the difference in TC counts between different climate states is significant, a bootstrap method was used to create 100 randomly selected 30-year samples out of the 40-year (1979 – 2018) distributions

of the annual number of observed TCs and ERA5 TCs.

## 3. Results

In this section, the TCs in the MH and PI climate conditions are studied to evaluate how changes in orbital forcing, dust and vegetation feedbacks impact TC activity in the Atlantic Ocean, by focusing on TC's trajectories and annual frequency (3.1), seasonality (3.2) and intensity (3.3). We also highlight the impacts of such changes on the different variables known to affect

TC genesis.

### 3.1 Change in TC Density and Frequency

The PI climate simulation has a spatial distribution of Atlantic hurricanes that is similar to present climate, where most of the TCs form in the Main Development Region (MDR) and move west-north-westward towards the North American East Coast (Fig. 2B). However, there are fewer TCs in the simulated PI climate than in the present-day climate simulation driven by

ERA-Interim (cf. Figs. 2A and A2); this is due to a large extent to the SST cold bias in EC-Earth simulation (Hazeleger et

al., 2012; Pausata et al., PNAS, 2017). When only the orbital forcing is considered (MH$_{PMIP}$), there is a northward shift of the Atlantic TC tracks, as well as an eastward displacement of the tracks away from the U.S. east coast at higher latitudes and a small increase in the TC track density relative to the PI experiment (Fig. 2C). This anomaly pattern is similar to that of the MH$_{GS+RD}$ experiment, but the anomalies are notably stronger in the latter simulation (Fig. 2D), extending further north and

westward into the Greater Antilles and Gulf of Mexico. The TC northward shift in the MH experiments and the strong eastward shift at higher latitudes are related to both the northward displacement of the ITCZ and the intensification of the WAM relative to the PI simulation (Fig. A3).

The northward shift of the ITCZ in the MH is due to energetic constrains associated with the changes in orbital forcing

causing a warming of the NH and a cooling of SH during boreal summer relative to PI (Merlis et al., 2013; Schneider et al., 2014; Seth et al. 2019). The ITCZ is associated with more favorable conditions for cyclogenesis by increasing the ambient vorticity and therefore the TC activity (Merlis et al., 2013). Our analysis shows that the absolute vorticity maximum undergoes a northward shift relative to the control experiment, following the ITCZ displacement (Fig. A4), supporting the northward shift of the TC tracks (Fig. 2C). Higher absolute vorticity values are also found over the Greater Antilles and the

western part of the Gulf of Mexico where there is a higher TC occurrence in the MH$_{GS+RD}$ relative to PI.

The northward shift and the increase of TC activity in the MH experiments is also related to the strengthening of the WAM, which amplifies the westerly winds – and the SST anomaly (Fig. A5). Such changes lead to the development of a wind shear pattern anomaly in the MDR, with lower values of wind shear in the central-western region of the MDR and higher values in

the eastern side of the MDR relative to PI. Thus, while the area more favorable for TC development is reduced (Fig. A6), the more favorable conditions present on the western side more than compensate the decrease in the east, allowing more cyclones to develop in the MH experiment. In addition to the zonal atmospheric circulation changes, the enhanced northward penetration of the WAM together with the displacement of the ITCZ leads to a northward shift of the maximum in African Easterly Waves (AEWs) activity in the MH experiments relative to PI (see Fig. A7). The poleward displacement of the

AEWs may also contribute to the changes in TC genesis location as they influence the region where TCs develop (e.g., Caron & Jones, 2012).

The vegetation changes and the associated reduction in dust concentrations further strengthen the WAM in the MH$_{GS+RD}$ relative to the MH$_{PMIP}$ experiment (Pausata et al., 2016; 2017), hence amplifying the changes seen in the MH$_{PMIP}$.

Furthermore, the reduction in dust concentration in the MH$_{GS+RD}$ experiment directly affects the SST (Fig. A5B), leading to an environment more prone to develop TCs relative to the MH$_{PMIP}$ and PI simulations. This is consistent with previous studies that found that the Sahara dust layer can have large impacts on TC activity (Evan et al., 2016; Pausata et al., 2017; Reed et al., 2019).





The GPI anomalies of both MH experiments relative to PI closely follow the changes in the atmospheric and oceanic
       environmental factors that can affect TCs (cf. Figs. 3, A4, A5, A6 and A7). The GPI shows more favorable conditions with
       higher values of vorticity and SST and lower wind shear values. Similarly to the absolute vorticity field (Fig. A4), the GPI
       shows a small northward shift relative to the control experiment, thus contributing to the poleward displacement of the TC
       genesis locations and therefore the the TC tracks (cf. Figs. 3 and Fig. 4).


       The largest changes in GPI are seen in the $MH_{GS+RD}$ experiment (Fig. 3B). The greening of the Sahara and the reduced dust
       concentrations over the Atlantic Ocean not only lead to higher potential for cyclogenesis in the MDR, but also extend the
       region westward towards the Caribbean where the model simulates a higher occurrence of TCs in this experiment relative to
       the PI (see Fig. 2C). Overall, the changes in cyclogenesis density for both MH experiment follow closely the changes in GPI
(cf. Figs. 3 and 4), suggesting that GPI is a good predictor of the TC activity changes, even in very different climate states.

       In terms of frequency, an average of 5.5 TCs per year is simulated in the PI experiment (Fig. 2B). This is ~45% less than the
       present-day climatology (~10 TCs per year; Landsea, 2014), which is likely due to a strong cold bias in the SST of the
       coupled model simulation (Pausata et al., 2017). Many high-resolution global models have similar biases in the Atlantic (e.g.
Shaevitz et al., 2014; Wing et al., 2019). The $MH_{PMIP}$ experiment shows a small increase (+9%; statistically significant) in
       the TC frequency relative to the PI, highlighting the minor impact of the orbital forcing alone on the number of Atlantic TCs
       (Fig. 2B). In the $MH_{GS+RD}$ simulation, more TCs are generated (7 per year) with a significant increase of around 1.5 TCs per
       year (+27%) relative to the PI experiment (Fig. 2B). Bootstrap tests with both HURDAT2 (Fig. 5A) and ERA5 (Fig. 5B)
       datasets show that the chances of obtaining an increase of 27% (9%) in the mean of each distribution are significantly
(slightly) higher than the 95[th] percentile of these distributions. Our sensitivity experiments hence roughly show that the
       orbital forcing alone contributes for about 33% (~0.5 TC per year) of the total increase in TC frequency occurring in the
       $MH_{GS+RD}$ relative to the PI experiment, while the Sahara greening and reduced dust concentrations account for about 66% of
       this increase (~1 TC per year). Thus, these results suggest that the TC activity is strongly dominated by the vegetation and
       dust changes, in close agreement with Pausata et al. (2017).

**3.2 Changes in TC Seasonal Cycle**

       To analyze changes in TC seasonal cycle we consider changes in the monthly number of TCs, rather than change of the
       length of the TC season. The PI climate has a TC seasonal cycle that is similar to the present climate, with a peak in TC in
       September (Fig. 6). The MH experiments show a distinct pattern: a decrease in TC activity at the beginning of the hurricane
       season for both MH experiments (statistically significant for the $MH_{PMIP}$ in July and August; non-significant for the
$MH_{GS+RD}$), followed by a large increase at the end of TC season (statistically significant during September and October in the
       $MH_{PMIP}$; $MH_{GS+RD}$ from September to November, SON) relative to PI.





Gaetani et al. (2017), using the same global model experiments performed with EC-Earth, showed a large decrease in the AEWs in the $MH_{GS+RD}$ relative to the $MH_{PMIP}$ simulation due to the strengthening of the WAM. As AEWs can potentially act

as seeds for TC genesis (Caron & Jones, 2012; Frank & Roundy, 2006; Landsea, 1993; Thorncroft & Hodges, 2001), we analyze the changes in the AEW seasonality in the MH experiments relative to PI to determine whether there is a direct link between the changes in the seasonality of AEW and TCs (Fig. 7). The AEWs activity is remarkably reduced between July to September – 80% less relative to PI – and intensified in October and November in the $MH_{GS+RD}$ relative to PI (Fig. 7B).

The reduction of the AEWs in the $MH_{GS+RD}$ experiment is related to the strengthening and northward shift of the West African Monsoon (WAM). The anomalous westerly wind flow associated with the northward expansion of the WAM rainfall significantly alters the African Easterly Jet (AEJ) (Fig. 8) and the Sahara Heat Low. In particular, the disappearance of the 600 hPa AEJ and northward displacement of the wind circulation is responsible for the lower frequency of AEWs during the summer months in the $MH_{GS+RD}$ relative to PI. On the other hand, the withdrawal phase of the WAM towards the

end of the season (SON) is associated with an increase in the frequency of AEWs relative to PI, potentially contributing to the increase in TC activity in those months. While the changes in the frequency of AEWs in the $MH_{GS+RD}$ are potentially in agreement with the simulated changes in TC seasonality, the frequency of AEWs in the $MH_{PMIP}$ is higher relative to PI, especially in June and July, which is at odds with the changes in TC frequency in the $MH_{PMIP}$ experiment (no change in June and slightly decrease in July; Fig. 7A). Furthermore, in July and August fewer TCs are simulated in the $MH_{PMIP}$ relative to

the $MH_{GS+RD}$, which has fewer AEWs. Hence, it is not possible to draw a direct link between the changes in the seasonality of AEWs and TCs between the MH and the PI simulations. These results agree with the findings of Patricola et al. (2018) that, while AEWs can affect TC genesis, their contribution may not be necessary and TCs can also be formed from other processes. Furthermore, Vecchi et al. (2019) showed that a combination of the large-scale environmental factors (in particular ventilation) and the frequency of disturbances determined the TC frequency in their model.


Other factors could be playing a role in modifying the TC seasonal cycle. In particular, the shift in TC seasonal cycle could be related to changes in the orbital forcing, most importantly the precession of the equinoxes: during the MH the perihelium was in September instead of January as today, with the stronger insolation anomalies peaking in late summer at NH low latitudes. Furthermore, while higher potential intensity (due mostly to warmer SSTs, see Figs. A5 and A9) develops on the

western part of the MDR and most of the North Atlantic Ocean from June to September relative to the PI experiment, the strengthening of the WAM causes a cold anomaly response over the eastern part of the MDR, together with stronger vertical wind shear and weaker absolute vorticity values. The withdrawal of the WAM in late September then causes the decrease in wind shear, and positive anomalies in both absolute vorticity and SSTs to extend eastward. These environmental anomalies are likely the reason for the TC seasonality changes during the MH experiments (Fig. A9A, A10A, A11A). The cyclogenesis

anomalies and the GPI changes are consistent with these assumptions (cf. Figs. 9A and A12A).



Accounting for the Sahara greening and reduced airborne dust concentrations (MH$_{GS+RD}$) leads to even larger changes relative to PI (Fig. A9B, A10B, A11B), strengthening the GPI anomalies in the MDR (Fig. 9B). These changes strongly increased the total GPI over the ocean from September to November in the MH$_{GS+RD}$ experiment and lead to almost twice as

many cyclones in November relative to PI and MH$_{PMIP}$ experiments (see Figs. 6, 10 and Fig. A12B). Furthermore, there is a westward extension of the region prone to TC development towards the Greater Antilles and the Caribbean Sea from July to September relative to the other two experiments (Fig. 9B and Fig. 11), which is also reflected in the seasonal GPI (Fig. 3B). These anomalies led to a small increase in cyclogenesis over the Caribbean Sea relative to PI during this part of the season and may explain the increased TC activity during July and August in the MH$_{GS+RD}$ relative to the MH$_{PMIP}$ and why almost as

much cyclones formed during September (cf. Figs. 6, A12).

### 3.3 Changes in Intensity

To assess TC intensity, we considered the 10-m maximum wind speed in 3 h intervals and then classified them using the Saffir-Simpson scale categories. For the three experiments, most tropical cyclones reach only tropical storm or hurricane Category 1 (93%, 88% and 91% for PI, MH$_{PMIP}$ and MH$_{GS+RD}$, respectively) with only a few reaching Category 2 (7%, 11%

and 8%) (Fig. 12). Both MH simulations generated Category 3 hurricanes (~1% in both cases), while there are no major hurricanes in the PI experiment. Our analysis of the ACE also reveals that in general, the mean ACE per cyclone in the MH experiments is higher than in the PI experiment (~6.6x10$^{-4}$ m$^2$s$^{-2}$ and ~7.4x10$^{-4}$ m$^2$s$^{-2}$ for MH$_{PMIP}$ and MH$_{GS+RD}$, respectively, ~6.1x10$^{-4}$ m$^2$s$^{-2}$ for PI; see legend in Fig. 13C). The increase in ACE in MH simulations arises from two different aspect: (1) TCs in the MH climate are more intense than in the PI experiment (as shown in Fig. 12), therefore leading to higher rate of

energy generation, and (2) TCs in the MH experiments tend to last longer (PI: 199 h, MH$_{PMIP}$: 217 h and MH$_{GS+RD}$: 283 h; see legend Fig. 13C), meaning that the same amount of energy can be spend over a longer time lapse. The combination of these two aspects with increased mean TC count per season in the MH experiments (Fig. 2B and Fig. 13A) therefore leads to a larger total mean ACE per experiment in the MH simulations compared to PI (see legend Fig. 13B).

To better understand the cause of these changes, we turn to the seasonal V$_{PI}$ (Fig. 14) and examine the regions where the atmospheric conditions are more favorable for TC intensification. The area showing the most favorable conditions for cyclone intensification in the MH$_{PMIP}$ relative to the PI experiment is located around the central-western portion of MDR and extends northwards over the central Atlantic Ocean and westward along the northern most part of the US East Coast (Fig. 14A). Less favorable conditions are present east and south of the MDR where colder SSTs are present. The mean V$_{PI}$ pattern

for the MH$_{GS+RD}$ yields even stronger anomalies than the ones simulated by the MH$_{PMIP}$, with substantially more favorable conditions for intensification in the MDR (Fig. 14B). More conducive conditions are also present in the Caribbean Sea where markedly lower values of vertical wind shear are simulated (Fig. A6). The combination of more favorable environmental conditions (e.g. wind shear) along with the occurrence of more TCs crossing this areas in the MH$_{GS+RD}$ experiment relative to both PI and MH$_{PMIP}$ (see Fig. 2D) increases the chances of getting more intense and long-living cyclones. The main factors

contributing to the increase in $V_{PI}$ in the $MH_{GS+RD}$ relative to the PI and $MH_{PMIP}$ experiments are the warmer SSTs (~1.5°C higher; Fig. A5B) and enhanced levels of convective available potential energy (CAPE; Fig. A13) as direct consequence of the reduced dust emissions. In comparison to the change seen in the $MH_{PMIP}$ relative to PI, less favorable conditions for intensification are simulated north of the MDR in the $MH_{GS+RD}$ (Fig. 14B). Over all, the $V_{PI}$ anomalies for both MH experiments strongly resemble those presented in Pausata et al. (2017) and closely follow the changes in GPI, therefore

leading to more intense and potentially longer-living cyclones where better conditions are available for cyclogenesis.

## 4. Discussion and Conclusions

In this study, we use the regional climate model CRCM6 with a high horizontal resolution (0.11°) to better investigate the role played by vegetation cover in the Sahara and airborne dust on TC activity in the Atlantic Ocean during a warm climate period, the mid-Holocene (MH, 6,000 yrs BP). We compared two different MH experiments – where only orbital forcing is

considered ($MH_{PMIP}$) and where also changes in vegetation and dust concentration are accounted for ($MH_{GS+RD}$) – to a control pre-industrial experiment (PI).

The dynamical downscaling approach allows investigating whether the changes simulated in our study are consistent with those in Pausata et al. (2017), who used the same coupled global model simulation to drive a thermodynamical downscaling

technique (Emanuel et al., 2008) to assess changes in TC activity. Our results show that the Sahara greening and related reduction in dust concentrations ($MH_{GS+RD}$ experiment) significantly increase the number of TC in the North Atlantic Ocean by about 27%, whereas the increase associated with the orbital forcing alone is smaller (9%; $MH_{PMIP}$). In general, our results are consistent with the findings of Pausata et al. (2017), although the changes in TC activity between the MH experiments and the PI simulation are smaller. Furthermore, the displacement of TC activity is different in our study and most likely

related to the fact that dynamical changes in ITCZ and AEW are not accounted for in Pausata et al. (2017).

Gaetani et al. (2017) showed a strong decrease in AEW in the $MH_{GS+RD}$ simulations and suggested a potential impact in TCs activity; however, our analysis does not show a consistent relationship between the frequency of AEWs and tropical cyclones. These results support the findings of Patricola et al. (2018) who showed through a set of sensitivity experiments

that the AEWs may not be necessary for TC genesis. This is supported by observational studies that could not find a direct relation in the frequency of AEWs and TCs (Russell et al., 2017). Instead, the AEWs seem to play a more important role on the location of TC genesis rather than the total TC frequency. Furthermore, the RCM domain size and location of the lateral boundary conditions impacts the frequency of AEWs TCs inside the domain (Caron & Jones, 2012; Landman et al., 2005).

Our study suggests that the different orbital parameters together with the changes in the WAM intensity may have been the main causes of the changes in TC seasonality, offering better conditions for cyclogenesis towards the end of the hurricane

season. WAM intensity affects the wind shear in eastern side of the MDR. The WAM withdrawal towards the end of the summer extended the more favourable conditions from the central western portion towards the eastern portion of the MDR, causing an increase in TC activity during the second half of the season in the MH simulations. These results are consistent

with the findings of Korty et al. (2012) who also showed higher cyclogenesis potential towards the end of the PI hurricane season in their MH experiment, with likely increase in TC activity during October when the GPI is at its maximum. However, their results are based on the entire Northern Hemisphere while here we only focus on the North Atlantic Ocean. Our results also show that the GPI is able to represent the regions more prone to TC development in different climate states, in agreement with previous studies (Camargo et al., 2007; Koh & Brierley, 2015; Korty et al., 2012b; Pausata et al., 2017).

Finally, the reduced dust emissions in the $MH_{GS+RD}$ experiment induces an SST warming that enhances the available thermodynamic energy, increasing the VPI compared to the $MH_{PMIP}$ and PI experiments. This result supports the projections of more intense TCs in a warmer climate (Knutson et al., 2013; Walsh et al., 2016; Knutson et al., 2020).

In conclusion, our study highlights the importance of vegetation and dust changes in altering TC activity and calls for

additional modeling efforts to better assess their role on climate. Regional model simulations with atmosphere and ocean coupling could also be useful to better represent the interactions between TC activity and TC-ocean feedbacks as large amount of energy is transferred through TC activity between the atmosphere and the ocean (Scoccimarro et al., 2017). Additional paleotempestology records will be of paramount importance to validate model results. We strongly recommend the inclusion of both vegetation changes and dust feedbacks when investigating both past and future climates. In the view of

a potential future "regreening" of the Sahel or reduced Sahara dust layer, as shown in Biasutti (2013), Evan et al. (2016) and Giannini & Kaplan, (2019), our work suggests that these changes may enhance TC activity over the MDR, the Greater Antilles and the western portion of the Gulf of Mexico, and could generate more intense and potentially longer-living cyclones, increasing the vulnerability of society to damages from severe TCs.

**Appendix**

**Tracking algorithm**

In this study, we developed a tracking algorithm that makes use of a three-step procedure to detect cyclones, following previous studies (Gualdi et al., 2008; Scoccimarro et al., 2011; Walsh et al., 2007):

*Storm identification*

The storms are identified with the following criteria:

a.    The surface pressure at the center must be lower than 1013 hPa and lower than its surrounding grid boxes within a radius of 24 km ($2\Delta x$); this pressure is then taken as the center of the storm.



     b.    The center must be a closed pressure center so that the minimum pressure difference between the center and a circle of grid points in a small and a large radius around the center (200 km & 400 km radii) must be greater than 1 hPa and 2 hPa respectively.

c.    There must be a maximum relative vorticity at 850 hPa around the center (200 km radius) higher than $10^{-5}$ s$^{-1}$.

     d.    The maximum surface wind speed around the center (100 km radius) must be stronger than 8 m/s

     e.    To account for the warm core, temperature anomalies at 250, 500 and 700 hPa are calculated, where each anomaly is defined as the deviation from a spatial mean over a defined region. The sum of the temperature anomalies between the three levels must then be larger than 0.5 °C.

f.    If there are two centers nearby, they must be at least 250 km apart from each other, otherwise the stronger one is taken.

To identify the genesis and dissipative phases of the TCs, a double filtering approach was used, similar to that applied by Caron and Jones (2012). The aforementioned threshold values were used to first detect all the potential centers that could belong to a storm for each time step. Then, these criteria were enforced to the standard values (defined below) following the

literature (Gualdi et al., 2008; Scoccimarro et al., 2011; Walsh, 1997; Walsh et al., 2007), and these new threshold values were applied to each center to identify the ones that satisfied these enforced criteria among the potential weak ones pre-defined. The centers that satisfied the standard criteria were labeled by the algorithm as being strong centers (or real TC centers) while those who only satisfied the first set of criteria were identified as being weak centers (with standard values properties defined below).

The enforced criteria are the following:

     a.    Surface pressure at the center deeper than 995 hPa

     b.    Minimum pressure difference between the center and a 200 km and 400 km radius greater than 4 hPa and 6 hPa, respectively.

     c.    Relative vorticity maximum larger than $10^{-4}$s$^{-1}$

d.    Wind speed maximum above 17 m/s

     e.    Warm core temperature anomaly above 2°C

Another condition was added that only the strong centers needed to satisfy:

     f.    The maximum wind velocity at 850 hPa must be larger than the maximum wind velocity at 300 hPa.

In doing so, we avoided double counting cyclones that may decreased in intensity, before re-intensifying. Conditions e. and

f. are the main conditions that filtered the TCs centers from other low-pressure systems and extratropical cyclones, as TCs have a warm core in their upper part and stronger low-level wind speed than other storms.

*Storm tracking*

Storms were then tracked as follow: for each potential center found, the algorithm used the *nearest neighbor* method, also applied in many other studies (Blender et al., 1997; Blender & Schubert, 2000; Schubert et al., 1998), to find a corresponding

center in the following 3 h time interval within a 250 km radius around the storm center. Once two centers were paired, they





formed a storm track. The potential position of the next center to be a continuation of the storm was then calculated using the storm history, based on the position of the previous two centers, which allowed to establish a possible speed and direction for the predicted center. A similar procedure was applied in Sinclair (1997) and was derived from Murray & Simmonds (1991). The algorithm then searches around the last storm center using the nearest neighbor method and around this potential

position at the next time step to find a matching center. The nearest center was always chosen first.

*Storm lifetime*

Once a track was completed, it had to satisfy the last following conditions:

1. The TC had to exist for at least 36 hours (with a minimum of 12 centers at 3 h intervals)

2. The TC needed to have at least 12 strong centers along its entire track, so that the shortest TC had only strong
centers (36 h)

3. The TC had to travel at least 10 degrees (~1000 km) of combined longitude and latitude in its lifetime

4. The number of strong storm centers needed to represent at least 77% of a sub-part of the complete storm track delimited by the first and last strong center found by the algorithm. This way, we ensured that the storm was most of its time classified as a TC.






**Appendix Figures**

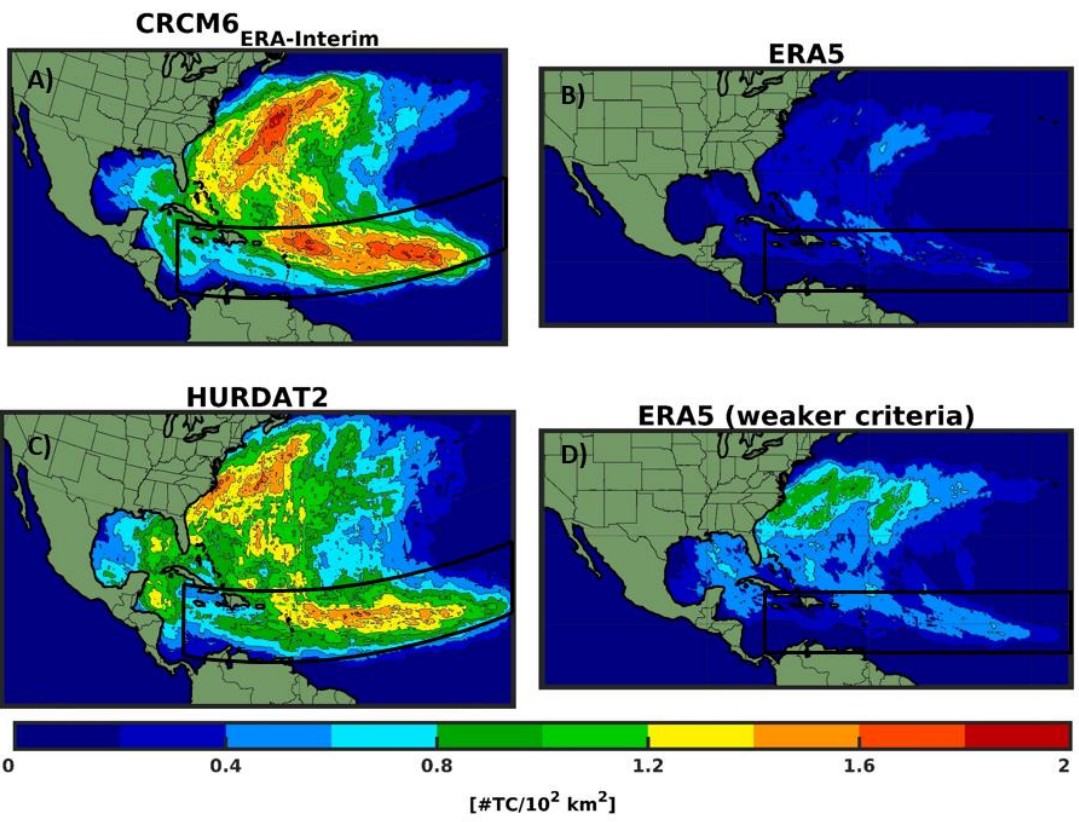

**Figure A1: June to November (JJASON) climatology (1980-2009) of (A) track density for the CRCM6 ERA-Interim driven experiment at 0.22°; B) ERA5 reanalysis data at 0.25°; C) Observed TCs from the HURDAT data base at 0.11° and D) ERA5 reanalysis data using weaker detection criteria at 0.25°. The black box shows the present-day Main Development Region (MDR). Note that the ERA5 figures are projected over a Mercator grid while the other two figures use the Equidistant Cylindrical Projection. The contour lines follow the colorbar scale.**






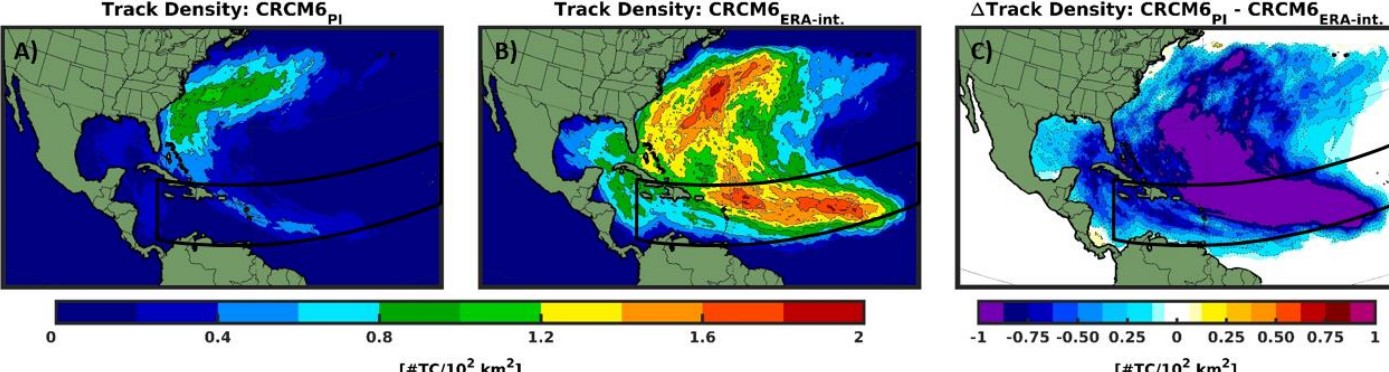

**Figure A2: Climatological track density (JJASON) for (A) pre-industrial experiment and (B) ERA-Interim driven experiment. (C) Changes in track density between the two simulations. The black box shows the approximate present-day Main Development Region (MDR). Only values that are significantly different at the 5% level using a local (grid-point) *t* test are shaded. The contour lines follow the colorbar scale (dashed, negative anomalies; solid, positive anomalies); the 0 line is omitted for clarity.**





**Figure A3: Monthly mean climatological changes in precipitation for (A) MH<sub>PMIP</sub> and (B) MH<sub>GS+RD</sub> relative to PI in the domain 23° W-75° W, 5° N-23° N. Only values that are significantly different at the 5% level using a local (grid-point) *t* test are shaded. The contour lines follow the colorbar scale (dashed, negative anomalies; solid, positive anomalies). The 0 line is omitted for clarity.**





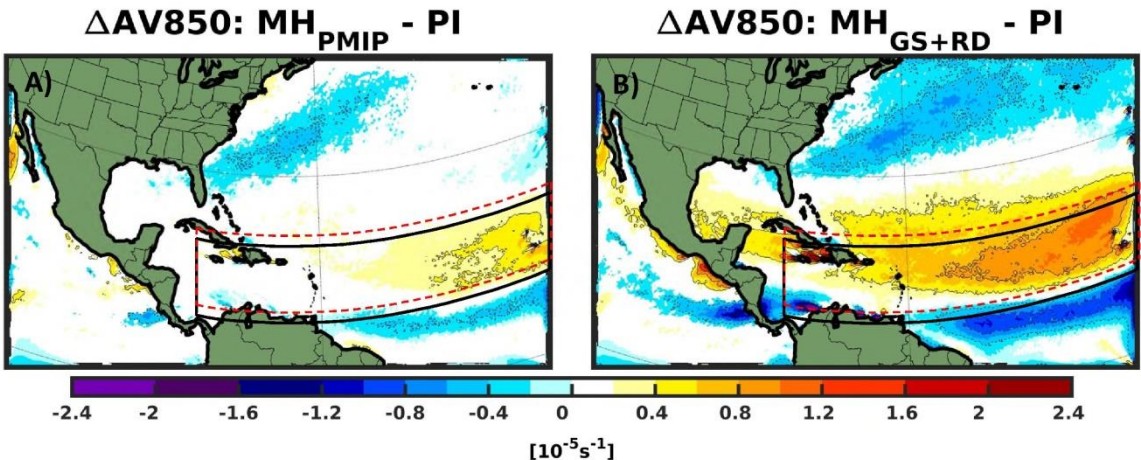

**Figure A4: Climatological (June to November, JJASON) changes in 850 hPa absolute vorticity relative to PI for (A) MH_{PMIP} experiment and (B) MH_{GS+RD} experiment. The black box represents the approximate present-day MDR. Red dotted box shows the approximate shift in the absolute vorticity maxima. Only values that are significantly different at the 5% level using a local (grid-point) _t_ test are shaded. The contour lines follow the colorbar scale (dashed, negative anomalies; solid, positive anomalies); the 0 line is omitted for clarity.**







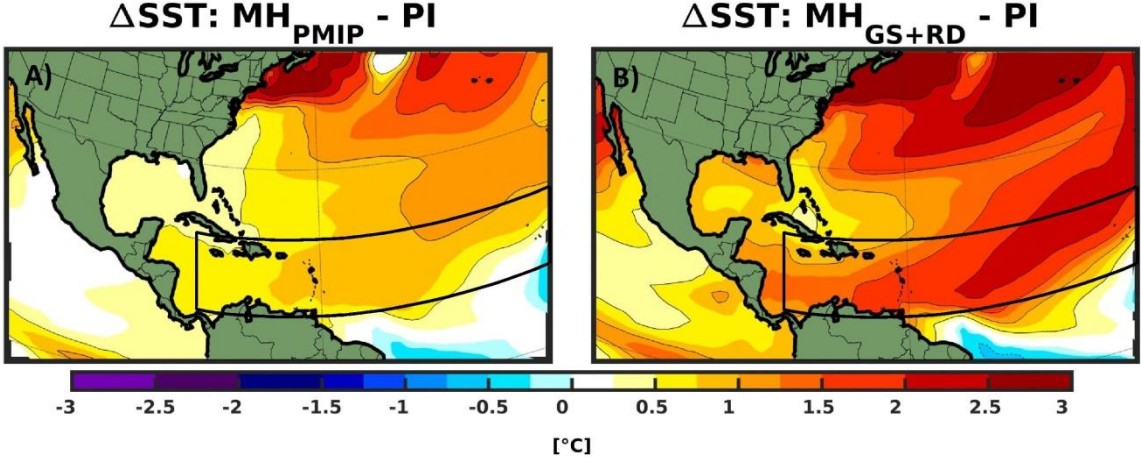

**Figure A5: Climatological (JJASON) changes in Sea Surface Temperature (SST) for (A) MH_PMIP and (B) MH_GS+RD experiments relative to PI. The black box shows the approximate present-day Main Development Region (MDR). Only values that are significantly different at the 5% level using a local (grid-point) $t$ test are shaded. The contour lines follow the colorbar scale (dashed, negative anomalies; solid, positive anomalies); the 0 line is omitted for clarity.**




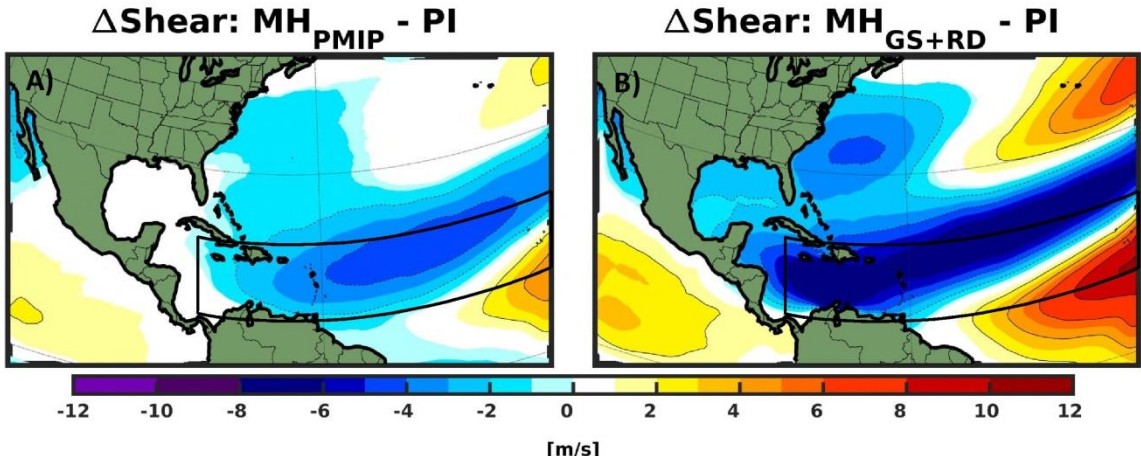

**Figure A6: Changes in vertical wind shear (300 hPa – 850 hPa) for (A) MH_{PMIP} and (B) MH_{GS+RD} experiments relative to PI. The black box shows the approximate present-day Main Development Region (MDR). Only values that are significantly different at the 5% level using a local (grid-point) *t* test are shaded. The contour lines follow the colorbar scale (dashed, negative anomalies; solid, positive anomalies); the 0 line is omitted for clarity.**






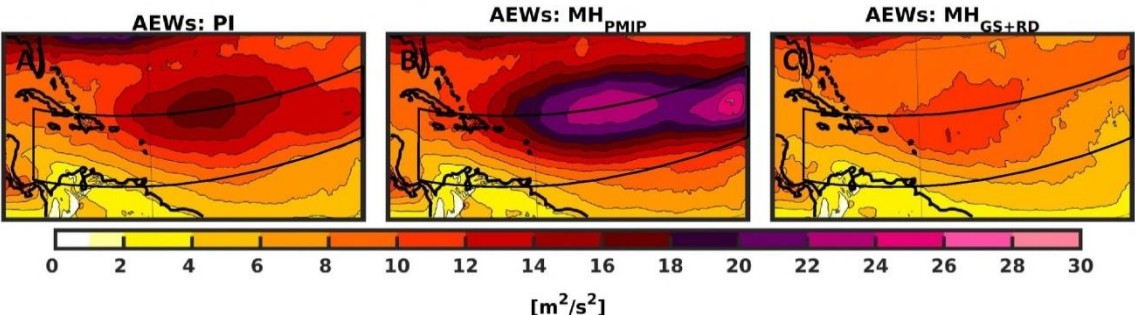

**Figure A7: African Easterly Waves represented through the variance of the meridional wind at 700 hPa, filtered in the 2.5- to 5-day band, for (A) PI, (B) MH_PMIP and (C) MH_GS+RD experiments. The black box shows the approximate present-day Main Development Region (MDR). The contour lines follow the colorbar scale.**






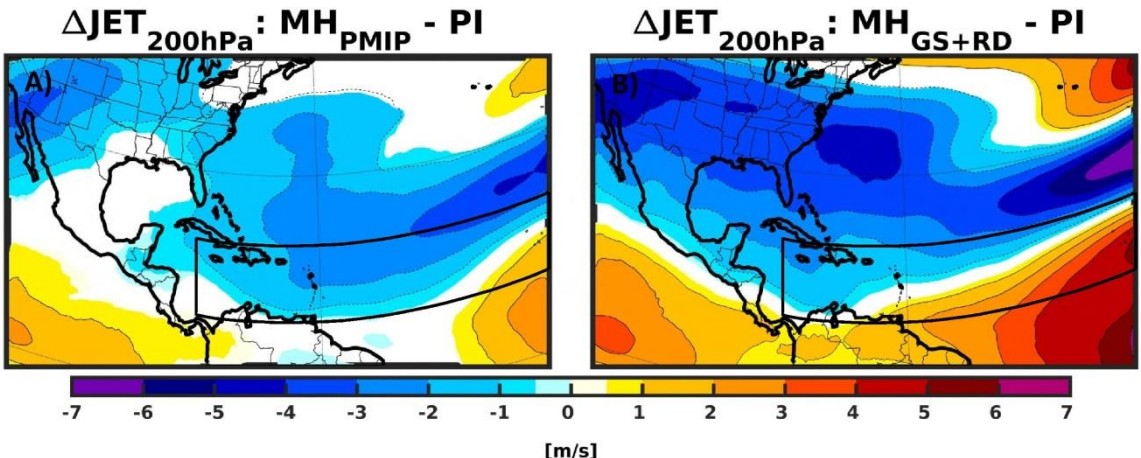

**Figure A8: Climatological (JJASON) changes in 200 hPa wind speed for (A) MH_PMIP and (B) MH_GS+RD experiments relative to PI. The black box shows the approximate present-day Main Development Region (MDR). Only values that are significantly different at the 5% level using a local (grid-point) *t* test are shaded. The contour lines follow the colorbar scale (dashed, negative anomalies; solid, positive anomalies); the 0 line is omitted for clarity.**





Figure A9: Monthly mean changes in climatological Sea Surface Temperature (SSTs) for (A) MH$_{PMIP}$ and (B) MH$_{GS+RD}$ relative to PI experiment. The black box shows the approximate present-day Main Development Region (MDR). Only values that are significantly different at the 5% level using a local (grid-point) $t$ test are shaded. The contour lines follow the colorbar scale (dashed, negative anomalies; solid, positive anomalies); the 0 line is omitted for clarity.





**Figure A10: Monthly mean changes in climatological wind shear (200 hPa – 850hPa) for (A) MH$_{PMIP}$ and (B) MH$_{GS+RD}$ relative to PI. The black box shows the approximate present-day Main Development Region (MDR). Only values that are significantly different at the 5% level using a local (grid-point) $t$ test are shaded. The contour lines follow the colorbar scale (dashed, negative anomalies; solid, positive anomalies); the 0 line is omitted for clarity.**



**Figure A11: Monthly mean changes in 850 hPa absolute vorticity for (A) MH$_{PMIP}$ and (B) MH$_{GS+RD}$ relative to PI. The black box shows the approximate present-day Main Development Region (MDR). Only values that are significantly different at the 5% level using a local (grid-point) *t* test are shaded. The contour lines follow the colorbar scale (dashed, negative anomalies; solid, positive anomalies); the 0 line is omitted for clarity.**





## A) ΔTC Cyclogenesis: MH_PMIP - PI

June

July

August

September

October

November

## B) ΔTC Cyclogenesis: MH_GS+RD - PI

June

July

August

September

October

November

**Figure A12: TC seasonal (June to November) cyclogenesis density anomaly for (A) MH_PMIP and (B) MH_GS+RD experiments relative to PI represented over a 5° meshgrid Mercator projection. The black box shows the approximate present-day Main Development Region (MDR).**

510



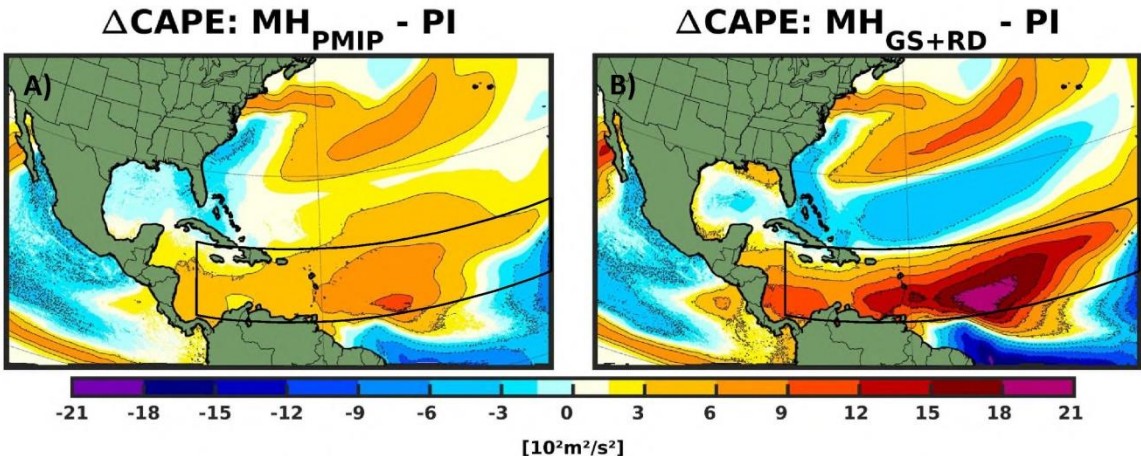

**Figure A13: Changes in climatological seasonal CAPE between a saturated boundary layer air parcel and an air parcel that has been isothermally lowered to a reference level (June to November, JJASON) for (A) MH$_{PMIP}$ and (B) MH$_{GS+RD}$ experiments relative to PI. The black box shows the approximate present-day main development region (MDR). Only values that are significantly different at the 5% level using a local (grid-point) *t* test are shaded. The contour lines follow the colorbar scale with different styles (dashed, negative anomalies; solid, positive anomalies); the 0 line is omitted for clarity.**

515



## Author contribution

F.S.R.P. conceived the study and designed the experiments with contributions from S.D. and R.L.. K.W. carried out the model simulations and S.D. analyzed the model output. S.D. and K.W. developed the tracking algorithm. S.J.C. and K.E. provided the codes to compute the indexes. All authors contributed to the interpretation of the results. S.D. wrote the manuscript with contributions from all co-authors.

## Competing interests

The authors declare that they have no conflict of interest.

## Acknowledgments

The authors would like to thank Georges Huard and Frédérik Toupin for the technical support, the Recherche en Prévision Numérique (RPN), the Meteorological Research Branch (MRB) and the Canadian Meteorological Centre (CMC) for the permission to use the GEM model as basis for our CRCM6 regional climate model, Qiong Zhang for sharing the global model outputs. This research was enabled in part by support provided by Calcul Québec (https://www.calculquebec.ca/) and Compute Canada (http://www.computecanada.ca). SD, FSRP and RL acknowledge the financial support from the Natural Sciences and Engineering Research Council of Canada (NSERC Grants RGPIN-2018-04981 and RGPIN-2018-04208). FSRP also acknowledge the financial support from the Fond de recherche du Québec - Nature et Technologies (FRQNT Grant 2020-NC-268559).

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





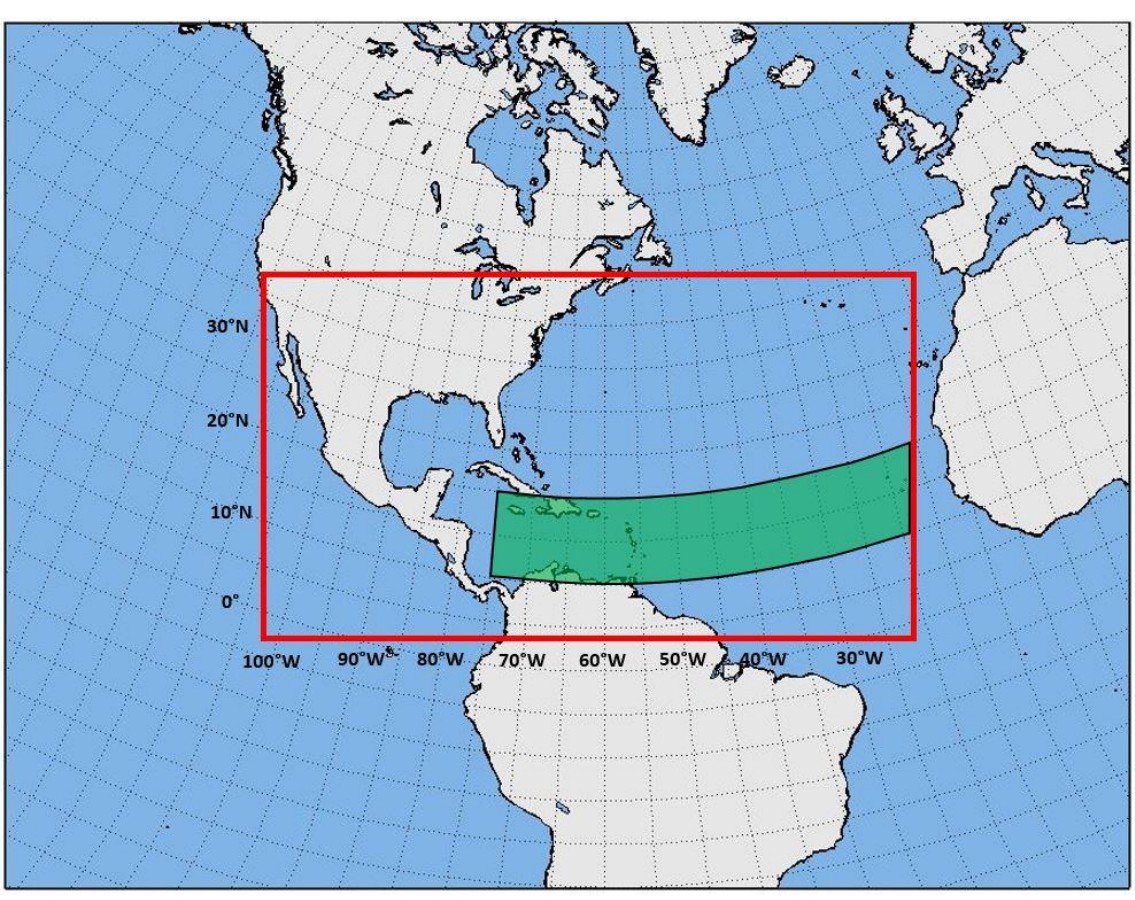

**Figure 1: CRCM6 simulation domain (red box). The black/green shaded box shows the approximate present-day Tropical Cyclones Main Development Region (MDR). Note that the data are projected over an Equidistant Cylindrical Projection.**






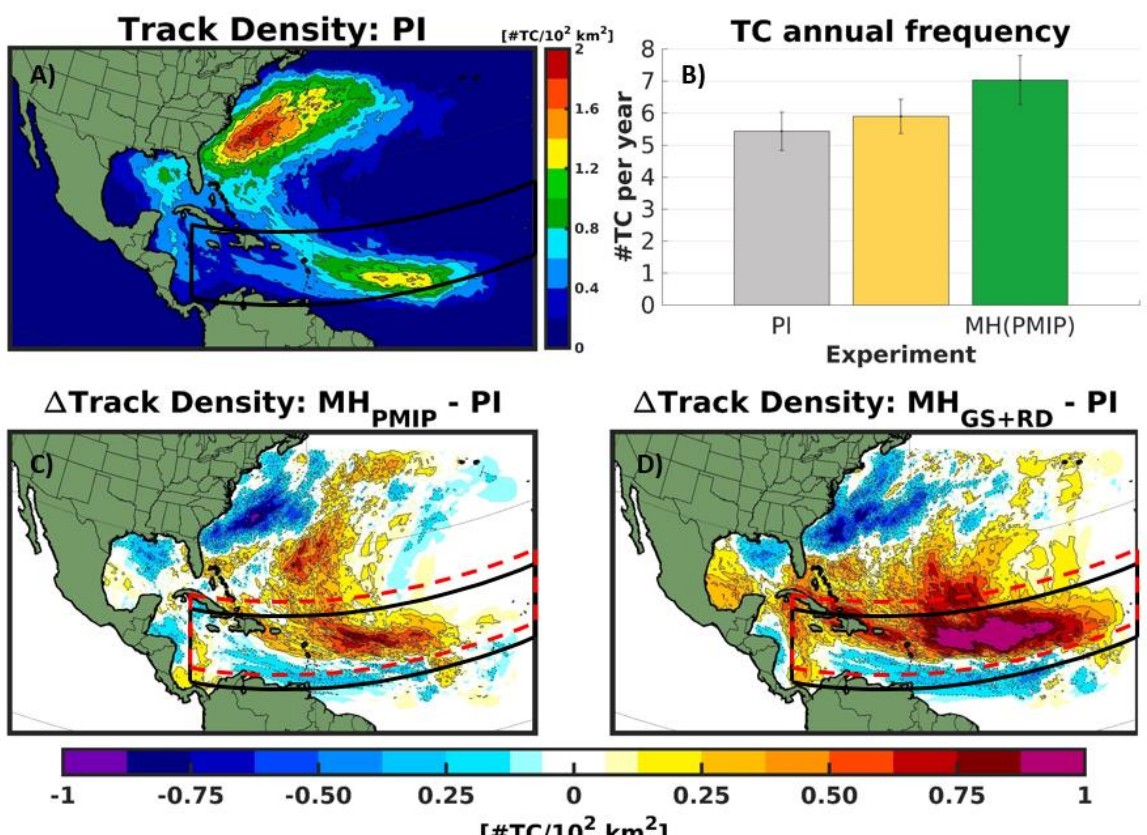

**Figure 2: June to November (JJASON) climatology of (A) track density for the preindustrial experiment (PI); (B) TC frequency in the Atlantic Ocean for each experiment. Error bars (whiskers) indicate the standard error of the mean; Changes in track density**
**for the MH$_{PMIP}$ (C) and the MH$_{GS+RD}$ (D) experiments relative to the PI. The black box shows the present-day main development region (MDR), the red dotted box shows the approximate shift of the MDR in the MH experiments. Only values that are significantly different at the 5% level using a local (grid-point) $t$ test are shaded. The contour lines follow the colorbar scale (dashed, negative anomalies; solid, positive anomalies); the 0 line is omitted for clarity.**






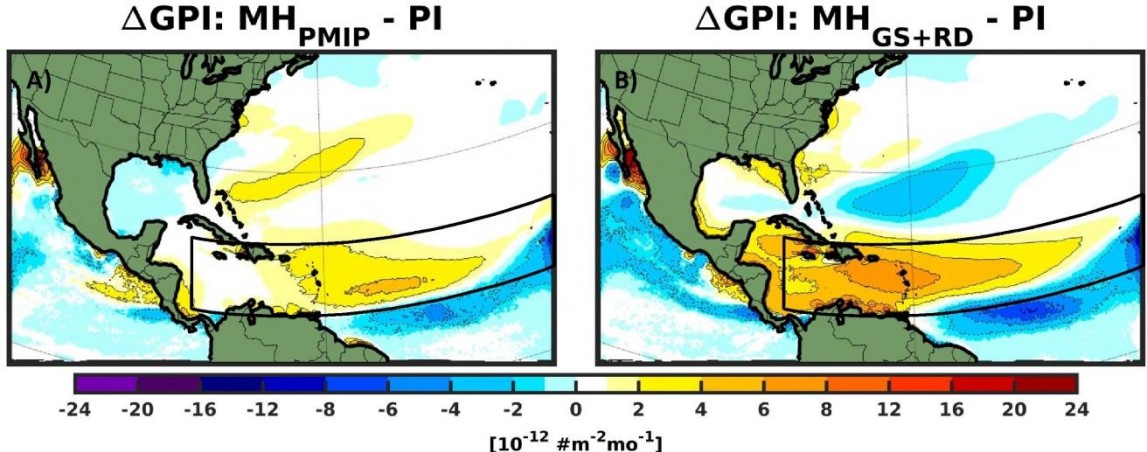

**Figure 3: Changes in seasonal Genesis Potential Index - GPI (June to November, JJASON) for (A) MH_PMIP and (B) MH_GS+RD experiments relative to PI. The black box shows the approximate present-day main development region (MDR). Only values that are significantly different at the 5% level using a local (grid-point) _t_ test are shaded. The contour lines follow the colorbar scale**

**(dashed, negative anomalies; solid, positive anomalies); the 0 line is omitted for clarity.**





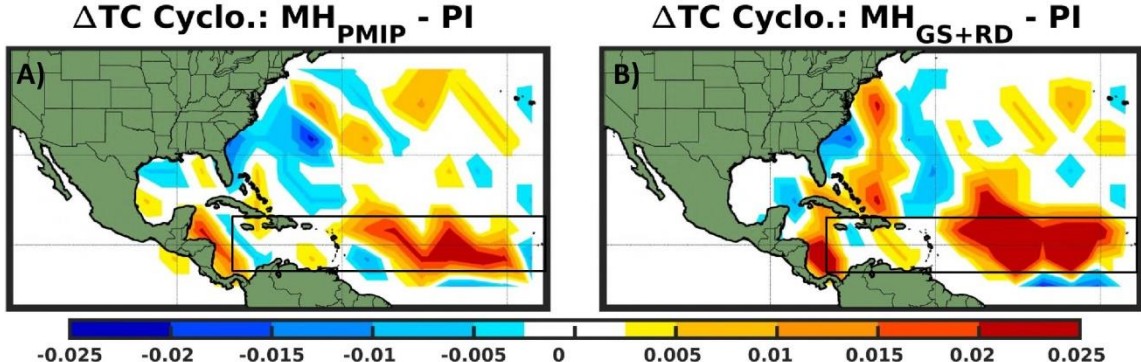

**Figure 4: TC seasonal (June to November) cyclogenesis density anomaly for (A) MH$_{PMIP}$ and (B) MH$_{GS+RD}$ experiments relative to PI represented over a 5° meshgrid Mercator projection. The black box shows the approximate present-day main development region (MDR).**





**Figure 5:** Bootstrap distributions based on (A) 1979-2018 HURDAT database and (B) 1979-2018 ERA5 reanalysis data. Legend
presents the median, mean, standard deviation, 5th and 95th percentiles of the distributions.





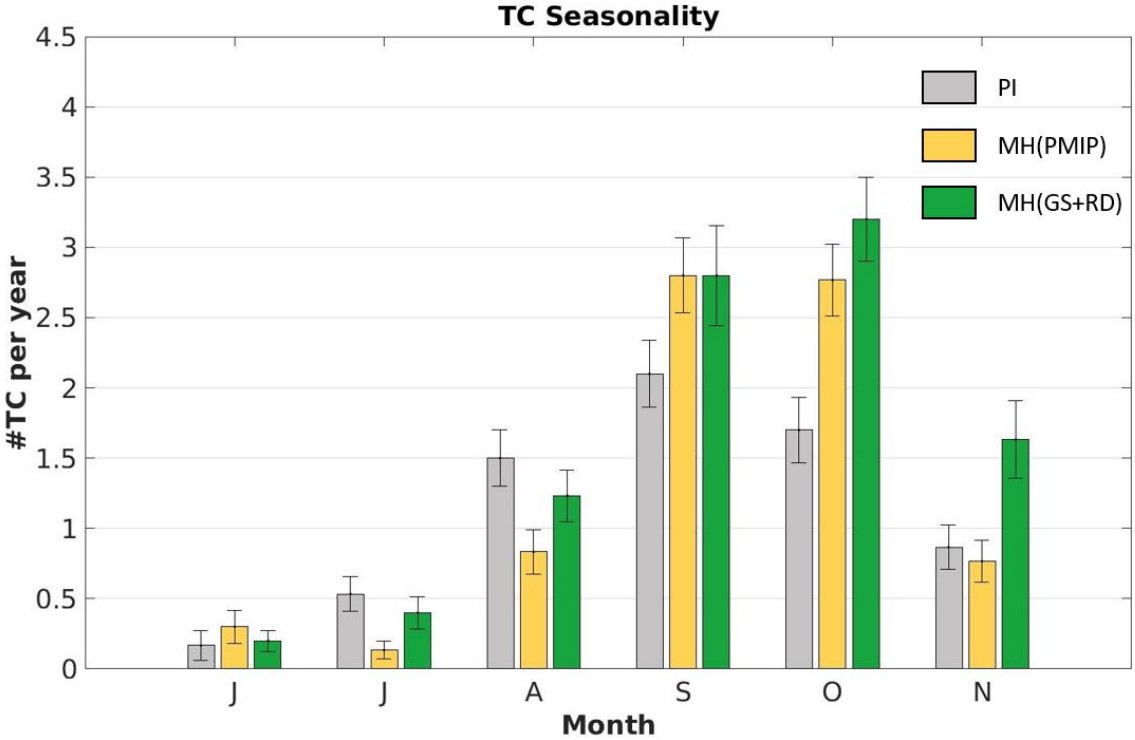

**Figure 6: TC climatological distribution throughout the extended TC season (June to November) for each experiment. Error bars (whiskers) indicate the standard error of the mean.**




**Figure 7: Monthly African Easterly Waves (AEWs) anomalies represented through the variance of the meridional wind at 700 hPa, filtered in the 2.5- to 5-day band, for (A) MH$_{PMIP}$ and (B) MH$_{GS+RD}$ relative to PI. The black box shows the approximate present-day main development region (MDR). Only values that are significantly different at the 5% level using a local (grid-point) *t* test are shaded. The contour lines follow the colorbar scale (dashed, negative anomalies; solid, positive anomalies); the 0 line is omitted for clarity.**





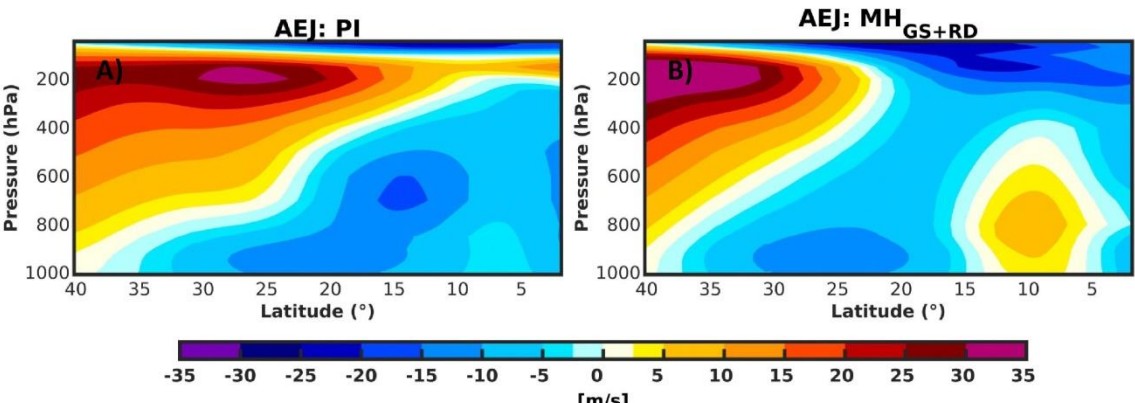

**Figure 8: African Easterly Jet (AEJ) represented through a vertical cross section of zonal mean (0-40°N; 20°W-30°W) seasonal climatological zonal winds for (A) PI and (B) MH_{GS+RD} experiments, respectively.**






**Figure 9: Changes in climatological monthly Genesis Potential Index (GPI) for A) MH_PMIP and B) MH_GS+RD relative to PI experiment. The black box shows the approximate present-day main development region (MDR). Only values that are significantly different at the 5% level using a local (grid-point) $t$ test are shaded. The contour lines follow the colorbar scale (dashed, negative anomalies; solid, positive anomalies); the 0 line is omitted for clarity.**





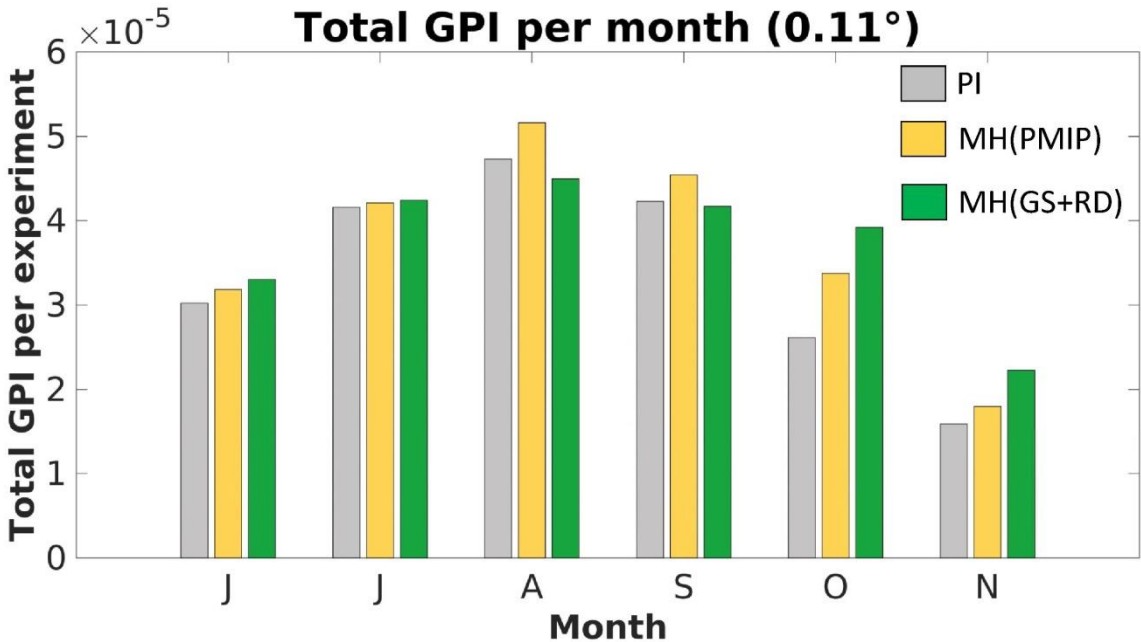

**Figure 10: Seasonal variation (June to November) of the Genesis Potential Index (GPI) summed over the experimental domain for the three experiments.**




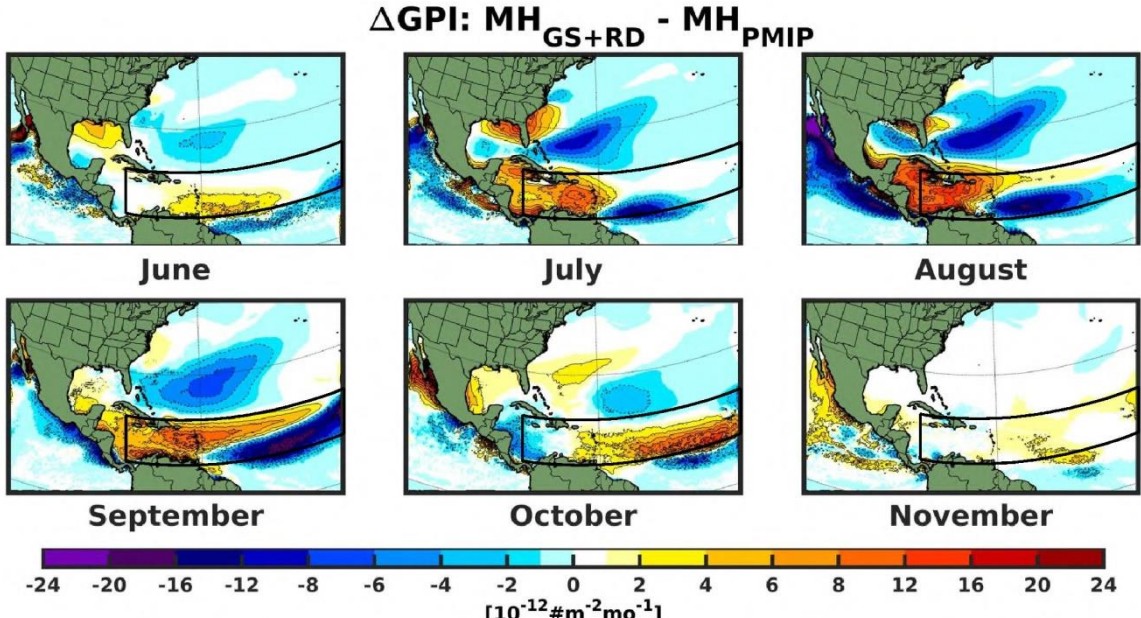

**Figure 11: Changes in climatological monthly Genesis Potential Index (GPI) for MH_GS+RD relative to MH_PMIP experiment. The black box shows the approximate present-day main development region (MDR). Only values that are significantly different at the 5% level using a local (grid-point) _t_ test are shaded. The contour lines follow the colorbar scale with different styles (dashed, negative anomalies; solid, positive anomalies); the 0 line is omitted for clarity.**





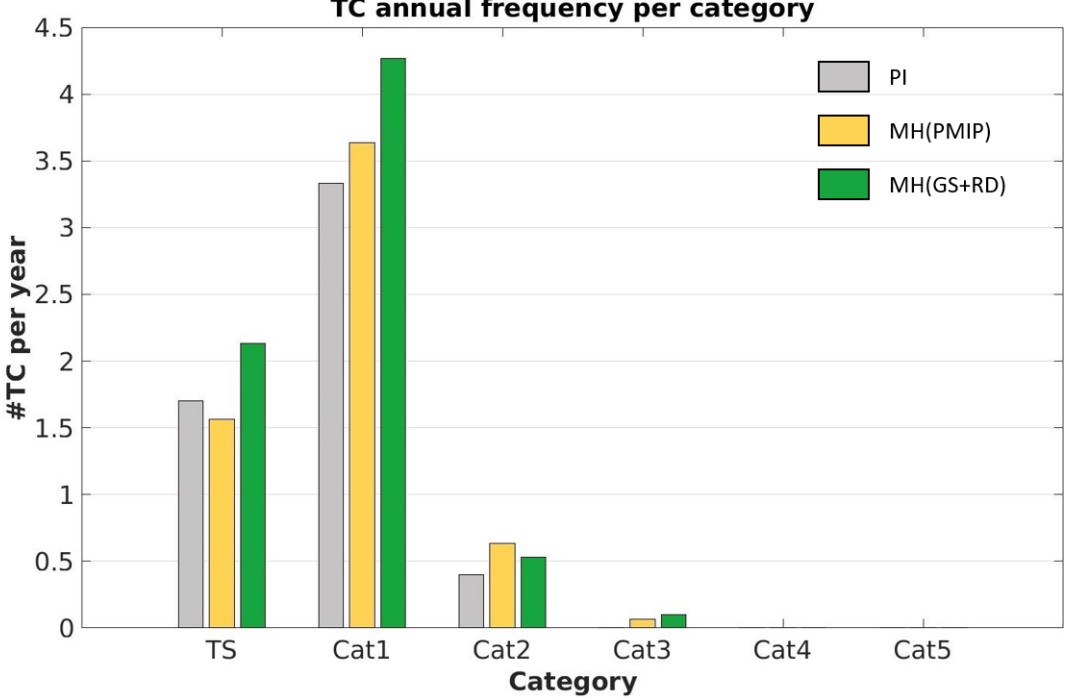

**Figure 12: Climatological number of TC per year in various categories for the three experiment during the TC extended season**
**(June to November; JJASON).**




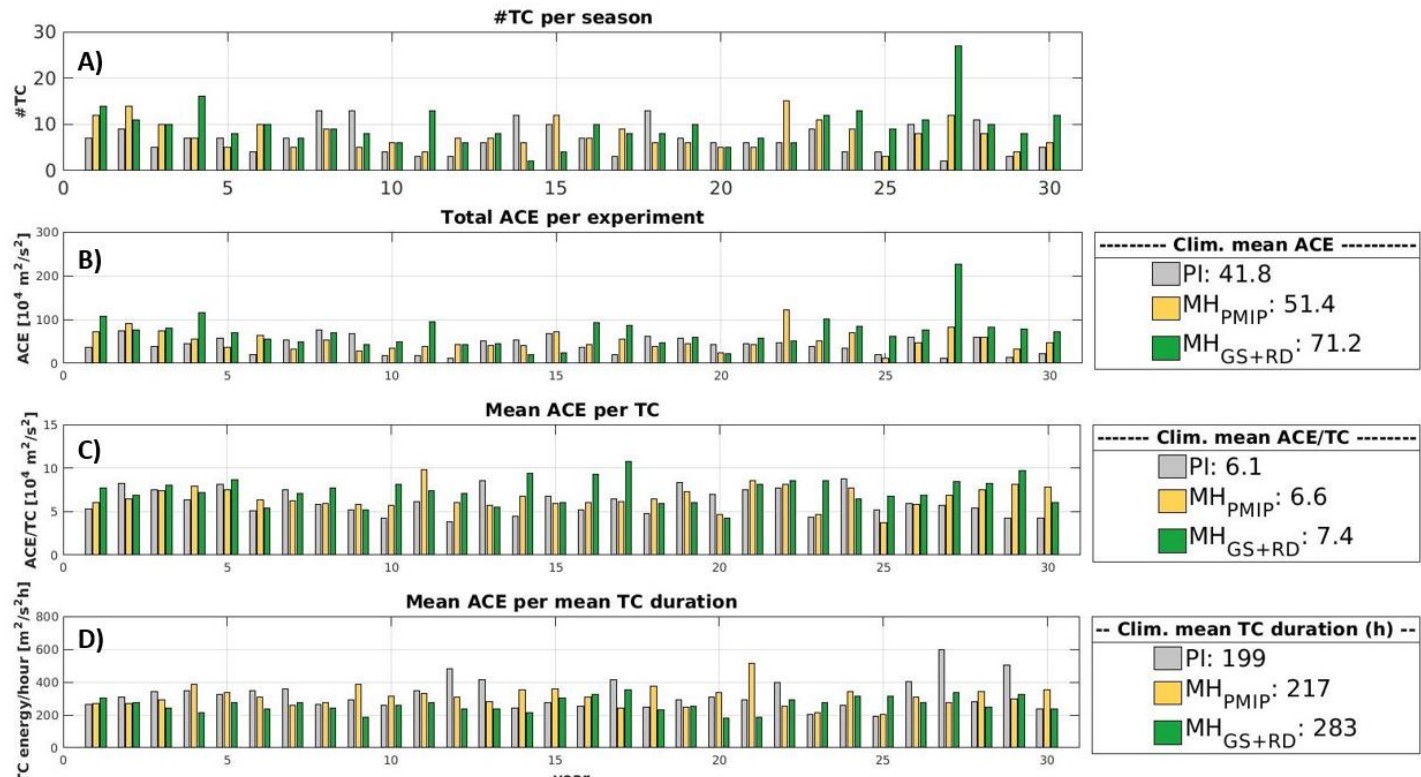

**Figure 13: (A)** Total number of cyclone per season for each experiment (30 years; Grey: PI, Yellow: MH_PMIP and Green: MH_GS+RD). **(B)** Total Accumulated Cyclone Energy (ACE; $10^4$ $m^2s^{-2}$). **(C)** Mean ACE per cyclone per season ($10^4$ $m^2s^{-2}$). **(D)** Mean ACE per cyclone normalized by the mean tropical cyclone duration in every season ($10^4$ $m^2s^{-2}h^{-1}$). Legends present the climatological mean of each distribution.



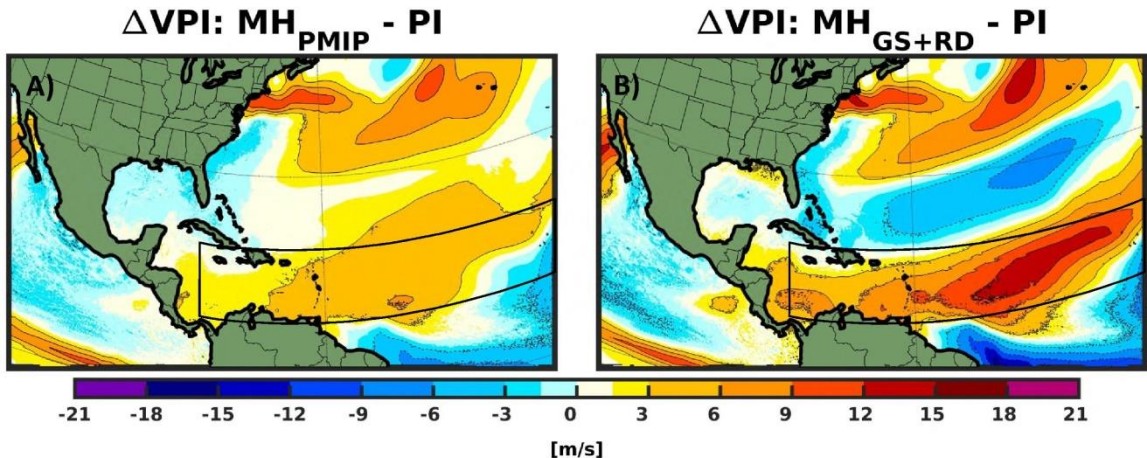

**Figure 14: Changes in climatological seasonal VPI (June to November, JJASON) for (A) MH$_{PMIP}$ and (B) MH$_{GS+RD}$ experiments relative to PI. The black box shows the present-day main development region (MDR). Only values that are significantly different at the 5% level using a local (grid-point) *t* test are shaded. The contour lines follow the colorbar scale (dashed, negative anomalies; solid, positive anomalies); the 0 line is omitted for clarity.**