# Peer review of "Atlantic Hurricane response to Sahara greening and reduced dust emissions during the mid-Holocene"

_Climate of the Past, 2020_

## Referee Comment (RC1) · Robert Korty (Referee) · 8 Nov 2020

This is a very nice and useful study on the interesting responses of TC activity to orbital forcing, dust, and North African land surface properties. Each of these has direct relevance to model responses in mid-Holocene simulations, and understanding how TCs respond to them are useful for broader questions of hurricanes and climate. The paper is well written and there are only a few specific comments on the text, which I post at the end of this review. I do have a few general comments that I expect can be easily addressed with minor revisions.

General comments:

[Figure]

I see an interesting consistency between the changes in the seasonal cycle reported here and those reported in earlier papers on environmental conditions at mid-Holocoene that I think you would be well-served to highlight. Korty et al. (2012), who studied the response to PMIP forcing (i.e., orbit only), reported that the Northern Hemisphere season was less favorable during its summer months, but that the environment switched to become more favorable by October and November (see their discussion at several points in their section 4 as well as their Figure 8). As discussed in that paper, the response was largely driven by the behavior of potential intensity (Vp): the atmosphere warmed faster in response to positive summer radiation anomalies than did the ocean surface, resulting in lower Vp during early summer, but Vp was higher during autumn after the ocean surface warming caught up. Koh and Brierley didn't find the increase in October/November conditions to be robust from model to model (their page 1445), but my impression from their paper was that there was a robust decline in the earlier summer months, which is again consistent with your findings here and with those of Korty et al. Thus, your results on the timing of changes in Northern Hemisphere TC activity show important consistency with earlier work on this subject. That is, there is a repression of either modeled activity or in the environmental conditions that support them during the summer months in all of these studies, which is later offset by increases in autumn activity in many of them. The difference across these investigations has been whether this autumnal increase would be large enough to only partially offsets the summer decline, cancel out changes, or whether it overwhelms the decreases from earlier months. You note this at line 360, but I think you should highlight this comparison in the introduction, as well as in Section 3.2 where much of this detail is presented. Your work, along with that of Pausata et al. (2017), has shown that there is an important (even dominant) role for dust related to the Green Sahara, but I think these similarities with the earlier work on PMIP simulations shows there is an imprint from the orbital changes present in all of them also.

Your analysis of the AEWs and comparison with the results of Patricola et al. (2018) is important. I wonder whether this could be taken one step further by commenting

further on differences in genesis locations between Pausata et al. (2017) and this study. With a reduced AEW production at 6ka, do you see differences in genesis locations between explicitly generated storms (this study) and statistically downscaled (Pausata et al.) that are as large as a 20th century control? If there are differences in genesis location, do these have repercussions in track densities downstream? Finally, what are the implications (if any) for paleotempestology sites in the western part of the basin?

Specific comments:

Abstract: You characterize the mid-Holocene as a warm climate state, but the warming is neither global or year-round. . .wouldn't it be better to characterize this as a change in the amplitude of the annual cycle? It is, however, a warmer state during the Northern Hemisphere summer, which is perhaps what you meant here, but if so you should clarify it explicitly. Hurricanes will respond to transient radiation anomalies differently than to warming from other sources, such as increases in CO2—e.g., Emanuel and Sobel 2013, for example—and readers should be cautioned about that.

General comment about the opening paragraph: This is a rather general introduction to why hurricane-climate studies matter, but I think it might strengthen this paper to start off focused with why studying the effects of dust emissions from the mid-Holocene is useful. I think your work makes a strong case that dust matters (perhaps even overwhelming the large orbital changes), so I recommend leading off with that. This has relevance to modern times as interannual Saharan dust variability can have large effects on Atlantic activity, and these effects are consequential for understanding what changes may come in the decades ahead.

Line 27: replace "rebuilt" with "rebuild". Also, "infrastructure" is more accurate than "vast quarters. . ."

Line 37: "recent decades" rather than "last decades" (the Emanuel paper cited was published 15 years ago, and it reported on storm trends primarily covering the decades

of the late 20th century)

Line 66: As you note, neither the Korty et al. nor Koh and Brierley papers studied simulated TCs, so it is not accurate to say they suggest a decrease in "activity". It would be better to say they found the orbit changes induce changes in Northern Hemisphere summer that—all else being equal—make environments more challenging for TCs.

Line 66: the meaning of "despite changes in summer insolation forcing" is unclear and I found the phrase is confusing. (Especially because you are talking about both Northern and Southern Hemispheres in this sentence, and insolation anomalies are opposite between them.)

Line 154: Just curious, at what specific pressure levels (or layers) do you calculate the "boundary layer" and "midtroposphere" entropies in the chi parameter?

Section 3.2: As I mentioned above, I think highlighting the similarities and differences between your results in this section with the predictions of Korty et al. and Koh and Brierley (based on orbit only) is important. There is an important similarity in that early summer months appear less conducive for TCs across all of these studies, with an increase later in the season in many of them. This suggests there is a detectable role of the orbit present in your results too, even if the Atlantic dust variations are strong enough to control the annual total.

---

## Referee Comment (RC2) · Anonymous Referee #2 · 19 Nov 2020

Review summary

This is well written and thorough study on hurricanes in the mid-Holocene. It should be published with minor corrections.

My main comment is about dust which is clearly a major factor in the results here. It is not clear that an ∼80% reduction in dust has the profound effect on the regional climate that is shown here and in the EC-Earth simulations (Pausata et al 2016). Most ESMs (including EC-Earth) are using OPAC dust measurements that are too absorbing (e.g. see discussion by Albani & Mahowald, 2019). This means that the radiative impact is likely strongly overestimated. Some discussion of caveats around this are therefore

needed.

General Comments

While the 80% reduction in dust is well established from mid-Holocene sediment core data, the radiative effect from this dust reduction is less obvious. Hopcroft & Valdes, 2019 showed that mid-Holocene dust reduction had a much smaller impact on the radiation balance in the HadGEM2-ES model. This is because HadGEM2-ES has more up-to-date physical dust properties which are significantly less absorbing than the OPAC dust data (Hess et al 1998) used in many ESMs. This means that the the strength of the modelled dust effect in the present study is probably too strong and this caveat should be discussed.

Also, related to this, Thompson et al 2019 showed that dust-cloud interactions can be important during the mid-Holocene. Would dust-cloud interactions impact the TC activity?

A general Climate of the Past reader may wonder how relevant the mid-Holocene can be for the future in terms of TC activity. I assume that the main effect in a future climate will relate to the warmer SSTs, whereas dust and WAM are secondary factors? Perhaps you can clarify this in the Discussion.

Specific Comments

Line 165: Do you mean in ERA5 itself? If Murakami and Hodges have questioned reanalysis why are you using it? Perhaps, this just needs some clarification?

Line 173: I think in this journal the units kt need to be explained.

Line 190: could you comment on the approximate magnitude of this SST bias?

Line 278: It may be worth noting some of these "other processes" here, or are these what is discussed in lines 281 onwards?

Line 336: It feels like a one or two sentence summary of your findings is missing here?

Line 344: "Furthermore, the displacement of TC activity is different in our study and most likely related to the fact that dynamical changes in ITCZ and AEW are not accounted for in Pausata et al. (2017)." I think this needs to be explained in a bit more detail.

Line 349: "These results support the findings of Patricola et al. (2018) who showed through a set of sensitivity experiments that the AEWs may not be necessary for TC genesis". Could you be a bit more specific?

Line 366: I'm not sure I agree with this statement. Surely, this is just a result from the model as are the projections for warmer climates? Maybe you mean that this warming-induced effect is consistent with the model-based projections for TC activity in a warmer future?

Line 373: "Additional paleotempestology records" - can you reference some?

Line 376: "our work suggests ..." yes but what about the surely much greater impact of SST warming from the $CO_2$ rise? Or is this less important?

Technical corrections

line 26 population -> populations Line 191: delete "PNAS". Line 199: constrains -> constraints line 371 "as large" -> "as a large"

References:

Albani, S and Mahowald, N. (2019). Paleodust Insights into Dust Impacts on Climate. J Climate, 32, 7897-7913, doi: 10.1175/JCLI-D-18-0742.1.

Hess, M., P. Koepke, and I. Schult, 1998: Optical properties of aerosols and clouds: The software package OPAC. Bull. Amer. Meteor. Soc., 79, 831–844, https://doi.org/10.1175/1520-0477(1998)079,0831:OPOAAC.2.0.CO;2.

Hopcroft, P., & Valdes, P. (2019). On the role of dust-climate feedbacks during the mid-Holocene. Geophysical Research Letters, 46 , 1612-1621. doi: 10.1029/

2018GL080483

Thompson, A., Skinner, C., Poulsen, C., & Zhu, J. (2019). Modulation of Mid-Holocene African Rainfall by Dust-Aerosol Direct and Indirect Effects. Geophys Res Lett , 46 , 3917-3926. doi: 10.1029/2018GL081225
* * *

---

## Author Comment (AC1) · 16 Jan 2021

See attached pdf document.

Please also note the supplement to this comment:
https://cp.copernicus.org/preprints/cp-2020-112/cp-2020-112-AC1-supplement.pdf
* * *